# Intrapulmonary T Cells Are Sufficient for *Schistosoma*-Induced Pulmonary Hypertension

**DOI:** 10.3390/ijms25179202

**Published:** 2024-08-24

**Authors:** Dara C. Fonseca Balladares, Biruk Kassa, Claudia Mickael, Rahul Kumar, Kevin Nolan, Thais C. F. Menezes, Michael H. Lee, Anthony M. Lau-Xiao, Ari B. Molofsky, Elina Wells, Brian B. Graham

**Affiliations:** 1Lung Biology Center, Division of Pulmonary and Critical Care Medicine, Zuckerberg San Francisco General Hospital, University of California San Francisco, San Francisco, CA 94110, USA; dara.fonsecaballadares@ucsf.edu (D.C.F.B.); biruk.kassa@ucsf.edu (B.K.); rahul.kumar2@ucsf.edu (R.K.); kevin.nolan@ucsf.edu (K.N.); michael.lee8@ucsf.edu (M.H.L.); anthony.lauxiao@ucsf.edu (A.M.L.-X.); 2Program in Translational Lung Research, Division of Pulmonary Sciences and Critical Care Medicine, University of Colorado Anschutz Medical Campus, Aurora, CO 80045, USA; claudia.mickael@cuanschutz.edu; 3Division of Respiratory Diseases, Department of Medicine, Federal University of São Paulo, São Paulo 04021-001, SP, Brazil; thais.menezes@unifesp.br; 4Department of Laboratory Medicine, University of California San Francisco, San Francisco, CA 94143, USA; ari.molofsky@ucsf.edu (A.B.M.); elina.wells@ucsf.edu (E.W.)

**Keywords:** pulmonary hypertension, schistosomiasis, CD4 T cells, FTY720

## Abstract

Background: Schistosomiasis is a parasitic infection that can cause pulmonary hypertension (PH). Th2 CD4 T cells are necessary for experimental *Schistosoma*-PH. However, if T cells migrate to the lung to initiate, the localized inflammation that drives vascular remodeling and PH is unknown. Methods: Mice were sensitized to *Schistosoma mansoni* eggs intraperitoneally and then challenged using tail vein injection. FTY720 was administered, which blocks lymphocyte egress from lymph nodes. T cells were quantified using flow cytometry, PH severity via heart catheterization, and cytokine concentration through ELISA. Results: FTY720 decreased T cells in the peripheral blood, and increased T cells in the mediastinal lymph nodes. However, FTY720 treatment resulted in no change in PH or type 2 inflammation severity in mice sensitized and challenged with *S. mansoni* eggs, and the number of memory and effector CD4 T cells in the lung parenchyma was also unchanged. Notably, intraperitoneal *Schistosoma* egg sensitization alone resulted in a significant increase in intravascular lymphocytes and T cells, including memory T cells, although there was no significant change in parenchymal cell density, IL-4 or IL-13 expression, or PH. Conclusion: Blocking T cell migration did not suppress PH following *Schistosoma* egg challenge. Memory CD4 T cells, located in the lung intravascular space following egg sensitization, appear sufficient to cause type 2 inflammation and PH.

## 1. Introduction

Pulmonary hypertension (PH) is a debilitating pulmonary vascular disease pathologically characterized by inflammation, yet the mechanisms underlying its pathogenesis remain elusive. While existing studies predominantly explore the role of innate immunity, the interplay between adaptive and innate immune responses that contribute to pulmonary vascular diseases are largely unexplored.

This research centers on *Schistosoma*, the causative agent of schistosomiasis, which causes the most prevalent form of Group 1 PH, also termed pulmonary arterial hypertension (PAH) [1], globally [2]. Our work employs a *Schistosoma*-triggered pulmonary hypertension (PH) murine model to shed light on the critical involvement of adaptive immunity in PAH. Prior work with this model identified the crucial role of Th2 activation of CD4 T cells in the lungs [3], resulting in the recruitment of CCR2^+^ Ly6C^+^ monocytes expressing thrombospondin-1 (TSP-1) to the adventitial space [4]. Functionally, TSP-1 activates latent TGF-β, modifying the phenotype of pulmonary vascular cells [4], and resulting in pulmonary vascular remodeling and PH. TGF-β signaling is shared with other forms of PH, including hypoxia, scleroderma, and idiopathic disease [5,6,7].

Here, we sought to understand the location-specific phenotypes and roles of CD4 T cells, and the potential requirement to migrate between locations, to unravel the mechanisms underlying *Schistosoma*-induced PH. The *Schistosoma*-PH mouse model employs intraperitoneal (IP) sensitization followed by intravenous (IV) challenge with *Schistosoma* eggs. A knowledge gap in this model is when CD4 T cells need to be located in the lung tissue to locally trigger type 2 inflammation as tissue-resident memory (TRM) Th2 cells or de novo circulating Th2 cells that migrate to the lungs.

CD4 T cells require the sphingosine-1-phosphate receptor (S1PR) to egress from lymph nodes. FTY720, also called fingolimod, is an S1PR functional antagonist approved by the Federal Drug Administration for the treatment of multiple sclerosis [8]. Administering FTY720 causes the arrest of T cells in the lymph nodes (LNs) and subsequent depletion of CD4 T cells in the peripheral blood [9,10], and has been used to specifically block Th2 cell trafficking to the lung [11,12]. Using FTY720, we tested key aspects of a *Schistosoma*-PH model, with a specific focus on the potential to block the recruitment of T cells into the lungs.

## 2. Results

### 2.1. Effect of FTY720 on T Cells in the Peripheral Blood and Mediastinal Lymph Nodes

To test the biological effect of FTY720 in *Schistosoma*-exposed mice, we assessed the density of CD4 T cells in the peripheral blood and mediastinal lymph nodes (MLNs) in the IP-sensitized and IV-challenged mice using flow cytometry. In the peripheral blood, we observed that FTY720 treatment starting immediately prior to IV egg challenge significantly decreased the fraction of lymphocytes (CD45+ cells) that were CD4+ or CD8+ T cells (Figure 1). In contrast, the percentage of lymphocytes that were T cells increased in the mediastinal lymph nodes (LNs) in FTY720-treated mice compared to vehicle-treated mice; the percentage increased significantly for CD3+ T cells, trended for the CD4+ and CD8+ T cells (Figure 2), and there were mild trends towards increased absolute numbers of T cells in the LNs. Note that a limitation of absolute number quantification is the potential difference in LN sampling between animals, and possible differences in LN mass or volume that may result from FTY720 treatment. Overall, these LN and peripheral blood data are consistent with the known effect of FTY720 arresting the efferent migration of T cells out of LNs.

### 2.2. Pulmonary Hypertension Phenotype

To investigate the location and phenotype-specific roles of CD4 T cells with respect to the PH phenotype, we characterized mice that were only IP sensitized to *Schistosoma* eggs, or IP sensitized and then IV challenged to *Schistosoma* eggs, with or without FTY720 treatment (Figure 3A). We observed that only IP sensitization did not cause a significant rise in the RVSP (Figure 3B), consistent with prior observations [13]. We found that IV challenge after IP sensitization did cause a rise in the RVSP, also consistent with prior data [3,4,13,14]. Unexpectedly, we observed no effect on the RVSP with FTY720 treatment starting prior to IV egg challenge. We assessed RV hypertrophy by measuring the mass of the RV free wall divided by the mass of the LV and septum (the Fulton index): this revealed no significant change in any group, which we attribute to the relatively short course of exposure at only 7 days after IV egg challenge (Figure 3C). The vascular remodeling phenotype was assessed by quantifying the fractional thickness of the medial layer of pulmonary vessels. This technique also found no change in media fractional thickness with FTY720 treatment (Figure 3D), corroborating the finding of no change in RVSP. IP egg sensitization alone did not cause vascular remodeling.

### 2.3. Pulmonary Inflammation Phenotype

Suppressing type 2 inflammation blocks *Schistosoma*-induced PH [3,14]. We did not find increased type 2 cytokine IL-4 or IL-13 concentrations following IP sensitization alone (Figure 4A,B), consistent with IP sensitization alone causing no significant PH phenotype. We found increased IL-4 and IL-13 concentrations with the addition of IV egg challenge, as had been previously observed [14]. However, the IL-4 and IL-13 levels were not modified using FTY720 treatment, paralleling the unchanged PH phenotype observed above.

We assessed the peri-egg granuloma volumes in the mice that had received IV eggs, as another indicator of *Schistosoma*-induced lung inflammation. Similar to the unchanged IL-4 and IL-13 concentrations, the peri-egg granuloma volumes were also not changed by FTY720 treatment (Figure 4C), further confirming that FTY720 treatment did not alter the degree of *Schistosoma*-induced lung inflammation.

### 2.4. Pulmonary T Cell Density and Phenotypes

We performed flow cytometry to identify the T cell numbers and phenotypes in the lung parenchyma, using the gating strategy shown in Figure 5. The intravenous administration of an anti-CD45 antibody prior to sacrifice was used to identify residual intravascular versus parenchymal cells [15]: the lung tissue was perfused with PBS prior to tissue harvesting, so the intravascular cells identified would likely be adherent to the vascular wall.

We unexpectedly observed in the mice that received only *Schistosoma* egg IP sensitization a significant increase in the intravascular density of CD45^+^ lymphocytes, CD4 and CD8 T cells, as compared to unexposed mice (Figure 6A–D). The density of intravascular lymphocytes and CD4 T cells then decreased back to the baseline after the IV egg challenge, whereas CD8 T cells continued to decrease. The addition of FTY720 treatment on top of IP sensitization and IV challenge resulted in unchanged lymphocyte density but trended towards decreased intravascular T cell density, consistent with the observed decrease in peripheral blood concentration, albeit that the adherent cells may be differently impacted as compared to non-adherent or circulating cells.

In contrast, the density of parenchymal lymphocytes and T cells did not significantly change with IP egg sensitization alone (Figure 6E–H). The addition of the IV egg challenge also did not significantly further impact the T cell density. However, FTY720 decreased the density, particularly of CD8 T cells, which was overall consistent with the concept that FTY720 suppresses the migration of T cells to the lung tissue.

The CD4 T cells were divided into three populations: naïve (CD62L^+^CD44^−^), memory (CD62L^−^CD44^+^), and effector memory/activated (CD62L^−^CD44^+^CD69^+^). We found a significant increase in the number of intravascular memory CD4 T cells in the lungs after IP sensitization alone, but no change in naïve or effector CD4 T cell density (Figure 7A–C).

IP sensitization alone did not significantly change the number of naïve, memory or effector memory CD4 T cells in the lung parenchyma, and the addition of the IV challenge also did not significantly change the density of these parenchymal T cells (Figure 7D–F). However, FTY720 treatment resulted in significantly fewer naïve CD4 T cells as compared to the control and IP only groups, and strongly trended towards fewer naïve CD4 T cells compared to the IP/IV plus vehicle group (*p* = 0.056).

Overall, the fraction of parenchymal CD4 T cells that were naïve decreased in the IP-sensitized and IV-challenged mice, and then further decreased with FTY720 treatment (Figure 7G).

## 3. Discussion

In this study, we investigated the effects of blocking lymphocyte egress from LNs, thus limiting blood pools available to traffic into the lung parenchyma in the *Schistosoma*-PH model, with the hypothesis that the development of PH requires the migration of CD4 T cells from mediastinal lymph nodes to the lung parenchyma following the IV embolization of eggs into the lungs [3], and that FTY720 would prevent PH by blocking the migration of those cells [16]. However, we found that the PH and inflammation severity remained unchanged by FTY720 treatment. FTY720 resulted in a significant decrease in naïve lung T cell density, but the fraction of effector CD4 T cells increased, such that the overall degree of inflammation remained unchanged.

Consistent with previous observations, IP sensitization alone did not induce a PH phenotype [3]. We observed an increase in the number of intravascular memory CD4 T cells in the lung after IP sensitization, consistent with a systemic inflammatory response, likely including the CD4 T cell binding. However, there was no significant change in the density of intraparenchymal T cells. These results suggest that even before IV egg administration, there were sufficient Th2 cells located in the lung vasculature to induce subsequent egg-associated, local type 2 inflammation.

Previous studies have demonstrated that CD4 T cells are important for orchestrating the type 2 inflammation that causes *Schistosoma*-induced PH [3,14,17,18]. There is also a requirement for CD4 T cells in hypoxia-induced PH [19,20]. Our observations in the *Schistosoma*-PH model may thus extend to other forms of experimental PH, such as with sterile inflammation induced through hypoxia.

In *Schistosoma*-PH, sensitization prior to IV egg challenge is required, presumably to educate naïve CD4 T cells, which then become further activated after the IV egg challenge and drive type 2 inflammation and PH [3,14]. A requirement for T cells was demonstrated by Rag knockout mice failing to develop PH after *Schistosoma* exposure [3]. Furthermore, the adoptive transfer of wild-type but not *Il4^−/−^Il13^−/−^* CD4 T cells to the *Rag^−/−^* mice restored the ability of the mice to develop *Schistosoma*-induced PH, whereas the adoptive transfer of CD4 T cells from sensitized wild-type mice to the *Rag^−/−^* mice enables the mice to develop PH after only IV *Schistosoma* egg challenge; together, these results indicate that Th2 CD4 T cells are both necessary and sufficient for *Schistosoma*-induced PH. In contrast, Th17 CD4 T cells are required in hypoxia-induced PH [19,20].

The current working model for how *Schistosoma*-PH is induced in mice begins with IP egg injection, resulting in antigen being taken up by peritoneal macrophages, which migrate to mediastinal LNs [21]. Within the LNs, macrophages transfer antigens to dendritic cells (DCs) [22], which locally present the *Schistosoma* antigen to naïve CD4 T cells. Those CD4 T cells with cognate T cell receptors are educated to the antigen and become memory CD4 T cells. Based on the results of our experiment here, it appears likely that some of these memory CD4 T cells leave the MLNs and migrate to the lung vasculature, even before the IV eggs are administered. Some CD4 T cells may preferentially cycle between lung tissue and mediastinal LNs, as it was reported that DCs can imprint lung-homing molecules on CD4^+^ T cells [23]. Subsequent IV egg administration results in the embolization of the ~50 µm diameter eggs to pre-capillary vessels in the lungs. The live eggs actively secrete antigens, which are taken up and locally presented to CD4 T cells by antigen-presenting cells in the lung tissue. This second exposure results in a reduction in intravascular T cells, presumably as they extravasate, and further results in memory CD4 T cells now becoming activated TEM with a Th2 phenotype, driving type 2 inflammation and PH.

The *Schistosoma*-PH model employs sequential IP egg sensitization and then IV egg challenge, which was historically used as a model of pulmonary type 2 granulomas [24]. Sensitization is crucial for sufficient immune activation following IV challenge with schistosome eggs [13]. Schistosome eggs in infected humans, however, will only be intravenously located: *Schistosoma mansoni* worms mate in portal veins, and their eggs embolize into the liver or through portocaval shunts to embed in pre-capillary lung vessels. The results of the FTY720 experiment suggest that functional differences between IP and IV sensitization are unlikely, as after IP sensitization alone intravascular memory CD4 T cells appear in the lung.

A robust Th2 inflammatory response occurs in *S. mansoni* infections as well as in response to other helminths [25]. Why *Schistosoma* is potentially unique as a parasitic driver of PH may be that the pathology includes extensive and chronic intravascular egg embolization into the lung, resulting in a progressive vascular disease that becomes irreversible [18,26]. The occurrence of schistosomiasis is related to the poverty rate and starts very early, affecting pre-school and school-aged children, impairing physical and intellectual development, and perpetuating the cycle of poverty [27,28,29,30]. Considering the global impact of parasitic helminths, understanding the pathophysiology and immunological process enables the development of effective therapies and prevention strategies, such as vaccines [16].

The requirement for T cell location in the host response to infections has been studied in other diseases using FTY720. In primary *N. brasiliensis* infection, the migration of CD4+ T cells is mandatory for host immunity, but the Th2 response to secondary infection is independent of both T cell migration and tertiary lymphoid structures in the lungs [16], observations similar to our findings here. Also consistent with our results, mice sensitized and then challenged with bacillus Calmette–Guérin (BCG) demonstrated that T cells in the lung were sufficient for the control of bacterial growth, even in the absence of newly recruited T cells from the lymph nodes [31]. Overall, while FTY720 is efficient at preventing the egress of naïve and TEM cells from lymph nodes to the lung, it does not impact TRM cells already present in the lung [16,31]. After an initial episode of allergen-induced asthma in an experimental murine model, memory CD4^+^ T cells also persist in the lung despite the administration of FTY720 [32].

A key strength of the current work is the mechanistic link between Th2 CD4 T cells, type 2 immunity, and PH severity. Limitations include the use of a pharmacologic modulator: aside from FTY720’s primary function to act as an S1PR antagonist to impact T cell migration from lymph node tissue, FTY720 can impact the function of other immune cells such as dendritic cells [33]. The effect of FTY720 inhibiting T cell migration to the lungs could only be partial, and a S1PR knockout animal was not tested. It is also possible that IP-administered antigens could be trafficked directly to the lung, where granulomas can serve as tertiary lymph nodes [34]. Some of the results suggested the clustering of the results (i.e., the IP/IV plus vehicle group in Figure 7E,F), although this is not consistently observed. There is likely some degree of experimental heterogeneity as a consequence of the tissue which was sampled, and the potential for egg clumping at the time of IV administration. The age of the mice was relatively young (6 to 8 weeks old at the start of the experiment; 9 to 11 weeks old at the conclusion), which has historically been used to induce a robust immune response; future experiments need to compare young versus aged mice to determine the phenotype in older age, which may be more relevant to human disease.

We also did not assess the density of innate immune cells, such as monocytes and macrophages, in the peripheral blood, LNs, or lung tissue. It is not expected that FTY720 would have a direct effect on these cells. It would be expected that FTY720 would have an indirect effect, in that type 2 inflammation driven by Th2 cells is required for monocyte recruitment to the lung tissue where they become interstitial macrophages and express thrombospondin-1 (TSP-1) which activates pathologic TGF-β; thus, FTY720 treatment likely results in a decreased density of TSP-1 expressing interstitial macrophages.

In summary, we observed FTY720 administration before the IV *Schistosoma* egg challenge did not suppress the subsequent type 2 immune response and PH phenotype, which is likely related to the memory T cells already present in the lung vasculature. Future experiments should investigate mechanisms by which T cells appear in the lung following IP sensitization alone, potentially through a combination of recruitment, local proliferation, and/or antigen trafficking to the lung. By broadening our understanding of these pathways and cellular drivers of *Schistosoma*-PH, we can identify novel therapeutic targets and develop more precise interventions aimed at mitigating the immune response associated with this debilitating disease.

## 4. Materials and Methods

Animal models. Six- to eight-week-old wild-type female C57BL6/J mice were purchased from Jackson Laboratories. Only female mice were employed, as we have previously observed a similar phenotype between male and female animals, and also that PH, including Schistosoma-PH, is more prevalent in females [35,36]. All animal experiments were approved by UCSF ICUAC.

Schistosoma-Induced PH Model. We have previously described that mice which undergo sensitization and challenge with Schistosoma eggs can develop experimental PH [3,4,14]. Briefly, S. mansoni eggs were obtained through the homogenizing and harvesting of livers from cercaria-infected Swiss Webster mice provided by NIAID Schistosomiasis Resource Center at the Biomedical Research Institute (BRI, Rockville MD, USA). Experimental mice were intraperitoneally (IP) sensitized with 240 Schistosoma eggs per gram body weight, and some mice two weeks later underwent intravenous (IV) challenge with 175 Schistosoma eggs per gram body weight. Control mice were unexposed to S. mansoni or underwent IP egg sensitization only.

FTY720 Treatment. FTY720 (SML0700, Sigma-Aldrich, Burlington, MA) was diluted in vehicle (PBS with 1% ethanol) to a concentration of 5 mg/mL. Daily FTY720 (0.5 mg/kg) or vehicle alone (the equivalent volume) IP administration was started the day prior to the IV egg challenge. Three days after the IV eggs, N = 6–7 mice per group were sacrificed, and the lungs were digested for flow cytometry (see below). Seven days after the IV eggs, the N = 6–7 mice per group underwent terminal right heart catheterization and tissue harvest.

Right heart catheterization and right ventricular hypertrophy measurement. Right ventricle systolic pressure (RVSP) was assessed in sedated mice, as previously performed [3,4,14]. We administered IP ketamine–xylazine to anesthetize the mice, followed by tracheostomy and then the initiation of mechanical ventilation. Through an open abdomen and diaphragm, a small hole was made in the RV free wall using a 30 ga needle, and then a 1 Fr pressure–volume catheter (PVR-1035, Millar AD Instruments, Houston, TX, USA) was placed directly into the right ventricle (RV). The process was subsequently carried out with the left ventricle (LV) to assess the LV intracavitary pressure. After flushing the lungs with PBS, the right lobes were sutured closed at the hilum, and the left lung was infused with 1% low melt agarose. The left lung tissue was placed on ice for the solidification of the agarose, and then in formalin for formalin fixation and paraffin embedding (FFPE) for histological assessment. In contrast, the lobes of the right lung were snap-frozen in liquid nitrogen for the protein. RV hypertrophy was determined by dividing RV mass by the sum of LV and septum mass (also known as the Fulton index).

Vascular remodeling assessment. FFPE lung tissue was immunostained for alpha-smooth muscle actin (SMA) at 1:200 dilution (using an anti-SMA antibody from Dako, Agilent, Santa Clara, CA, USA), as previously reported [3,4,14]. Images from stained slides were assessed using a Nikon Eclipse E800 microscope (Nikon, Melville, NY, USA) with a CCD camera (Photometrics, Tucson, AZ, USA). The vascular media thickness was quantified using digital image processing software (Image Pro Plus v4.5.1, Media Cybernetics, Bethesda, MD, USA).

Estimated granuloma volume assessment. Hematoxylin and eosin-stained FFPE slides were used to assess the peri-egg granulomas. Images were acquired of granulomas that surround a single egg. The estimates of peri-egg granuloma volumes were made with image processing software (Image Pro Plus v10) by using the egg as the center point, via the optical rotator stereological method [37].

Protein assessment and ELISA. Snap-frozen lung tissue from the right lobes were macerated and sonicated in RIPA buffer containing anti-proteases from EMD Millipore (Billerica MA, USA). The anti-proteases included the following: Protease Inhibitor Cocktail Set I (I 539131-10VL), Phosphatase Inhibitor Cocktail Set II (524625-1SET), and phenylmethylsulfonyl fluoride (PMSF). All protease inhibitors were used at a 1:100 dilution. The total protein concentration in the whole-lung lysate was quantified using the Bradford assay (5000201, BioRad, Hercules, CA, USA). IL-4 and IL-13 concentrations in the mouse lung lysates were quantified through ELISA using kits from R&D systems (Minneapolis, MN, USA) (IL-4: M4000B-1, DY404-05, and DY008B; IL-13: M1300CB, DY413-05 and DY008B).

Flow cytometry assessment. Lungs, blood, and lymph nodes (LNs) from unexposed and Schistosoma-exposed mice were analyzed to characterize lymphocytes and CD4 T cells using flow cytometry, as described previously (3,4). To distinguish interstitial from circulating cells, BV570-labeled anti-CD45 antibodies (2 µg/mouse) were retro-orbitally injected into the mice to label intravascular leukocytes, with the injection occurring 5 min before euthanasia. Blood samples were collected first, followed by perfusing the lungs with PBS via instillation into the right ventricle. The PBS-flushed lungs were then collected and subject to digestion with liberase enzyme (Roche, Germany) dissolved in RPMI medium (Mediatech, Corning, NY, USA) at 37 °C for 30 min using a concentration of 1 mg/mL. Mediastinal LNs were also harvested and mechanically disrupted by passing the tissue through 16 ga and then 18 ga needles five times each, then followed by filtration through a 100 μm cell strainer (Fisher Scientific) and centrifugation. After tissue collection, red blood cells (RBCs) were lysed using 1 mL volume of ACK lysis buffer (Gibco). Then, the cells were resuspended and washed in RPMI. The resulting single-cell suspensions were filtered and then collected into flow wash buffer (5% BSA in PBS with EDTA) for subsequent staining for flow cytometry.

Blood and the digested lung and LN single-cell suspensions were pre-incubated with an anti-CD16/CD32 antibody for a 20 min duration to block non-specific Fcγ receptor-mediated antibody binding (see Table 1). Subsequently, the cells were stained with fluorochrome-conjugated antibodies at 4 °C for 30 min. Details regarding clones and concentrations of the antibodies used are provided in Table 1. The cell suspensions were analyzed with an LSRII (BD Biosciences) flow cytometer at the UCSF Core Immunology Laboratory (CIL). FlowJo (v-10.9.0) was used for analysis.

## Figures and Tables

**Figure 1 ijms-25-09202-f001:**
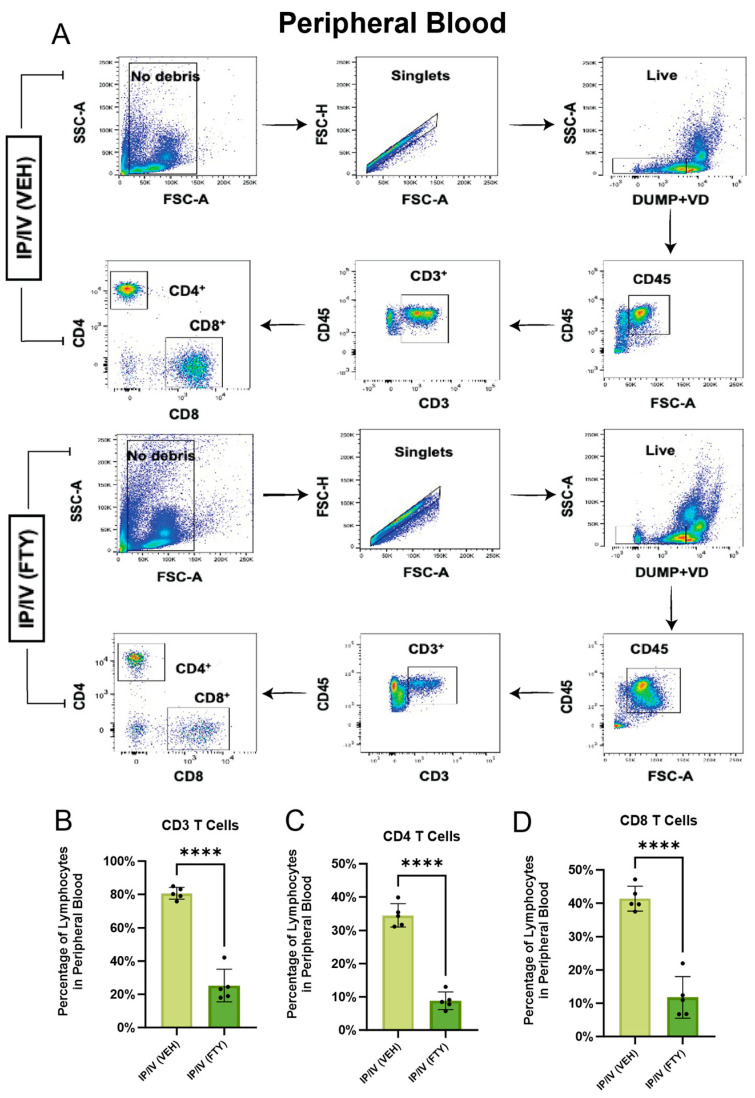
FTY-720 decreased T cells in peripheral blood, as expected. (**A**) Representative flow cytometry gating strategy for each condition. Fraction of CD45+ lymphocytes in peripheral blood, which are (**B**) CD3+ T cells, (**C**) CD4+ T cells, and (**D**) CD8+ T cells. Mean +/− SD plotted; N = 5/group; *t*-test *p* values shown; **** *p* < 0.0001.

**Figure 2 ijms-25-09202-f002:**
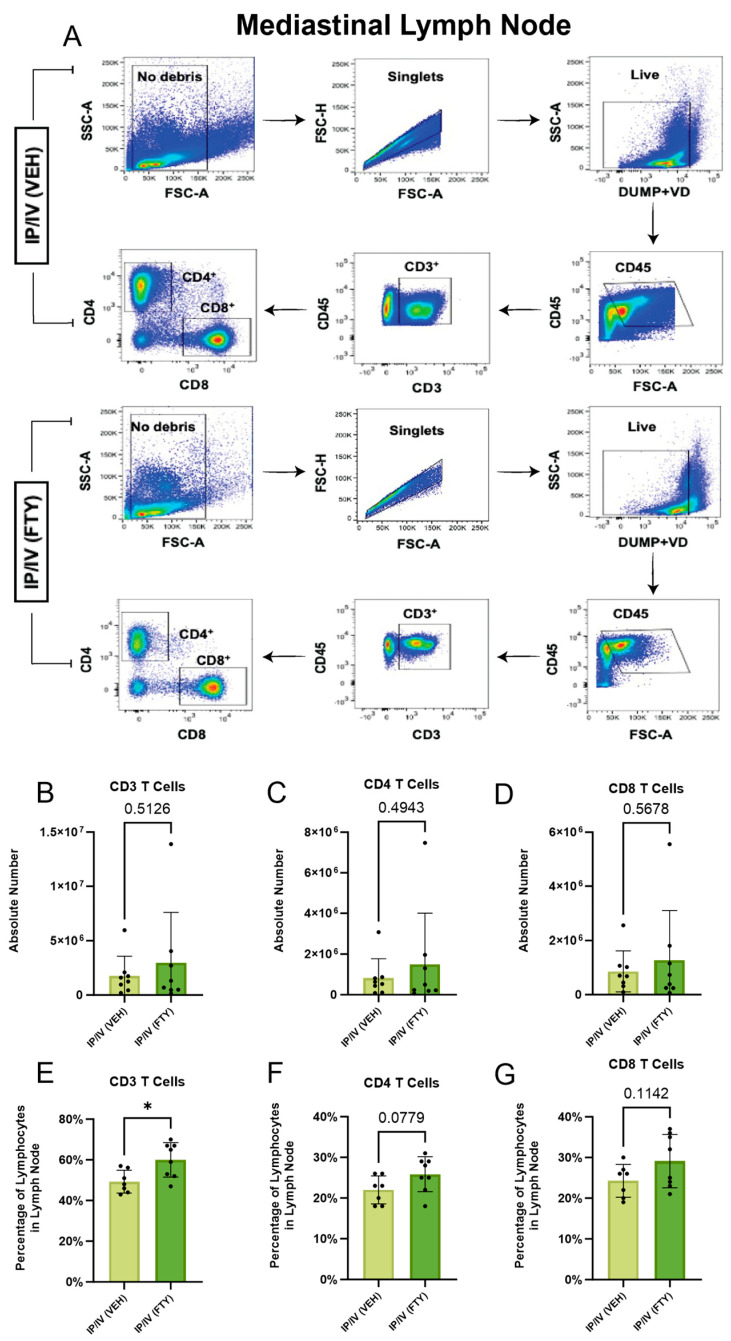
FTY-720 increased T cells in mediastinal lymph nodes, as expected. (**A**) Representative flow cytometry gating strategy for each condition. Absolute number per lymph node (**B**–**D**) and relative fraction of CD45+ lymphocytes (**E**–**G**) in peripheral blood, which are (**B**,**E**) CD3+ T cells, (**C**,**F**) CD4+ T cells, and (**D**,**G**) CD8+ T cells. Mean +/− SD plotted; N = 7–8/group; *t*-test *p* values shown; * *p* < 0.05.

**Figure 3 ijms-25-09202-f003:**
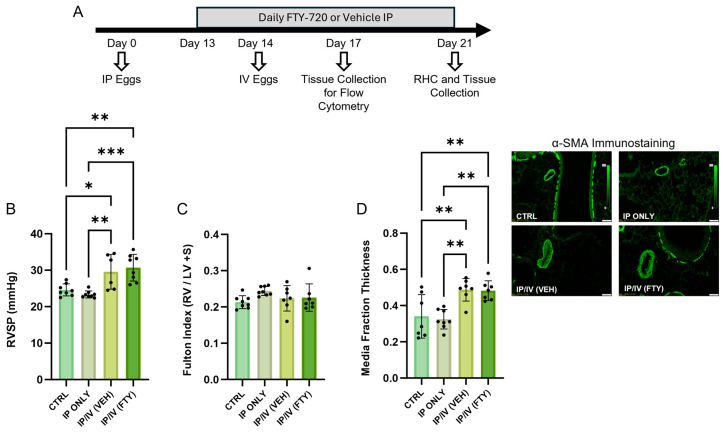
FTY720 treatment did not alter *Schistosoma*-PH severity, which is greater than in control animals. (**A**) Outline of experiment. (**B**) Right ventricle systolic pressure (RVSP), (**C**) Fulton index (mass of RV free wall divided by mass of LV plus septum), and (**D**) media thickness with representative examples in control, IP only, and vehicle (VEH)- and FTY720 (FTY)-treated mice IP sensitized and IV challenged with *S. mansoni* eggs. Mean +/− SD plotted; N = 6–8/group; ANOVA with post hoc Tukey testing *p* values shown; * *p* < 0.05, ** *p* < 0.01, ****p* < 0.001. Scale bars: 50 µm.

**Figure 4 ijms-25-09202-f004:**
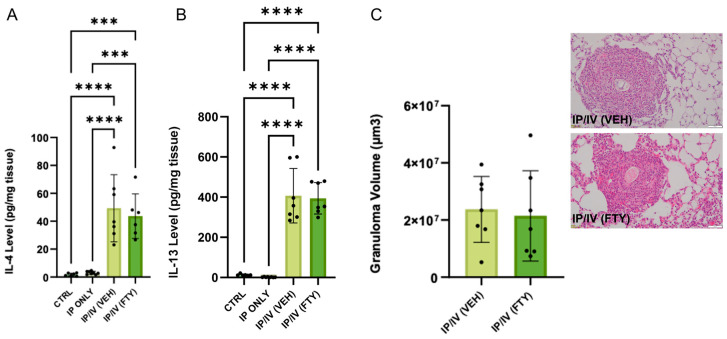
Lung inflammation severity was not altered using FTY720 treatment, which is greater than in control animals. (**A**) IL-4 and (**B**) IL-13 concentration, and (**C**) peri-egg granuloma volumes with representative images in control, IP only, and IP/IV *Schistosoma* egg-exposed mice treated with vehicle (VEH) or FTY-720 (FTY). Mean +/− SD plotted; N = 6–8/group; ANOVA with post hoc Tukey testing *p* values shown; *** *p* < 0.001, **** *p* < 0.0001. Scale bars: 100 µm.

**Figure 5 ijms-25-09202-f005:**
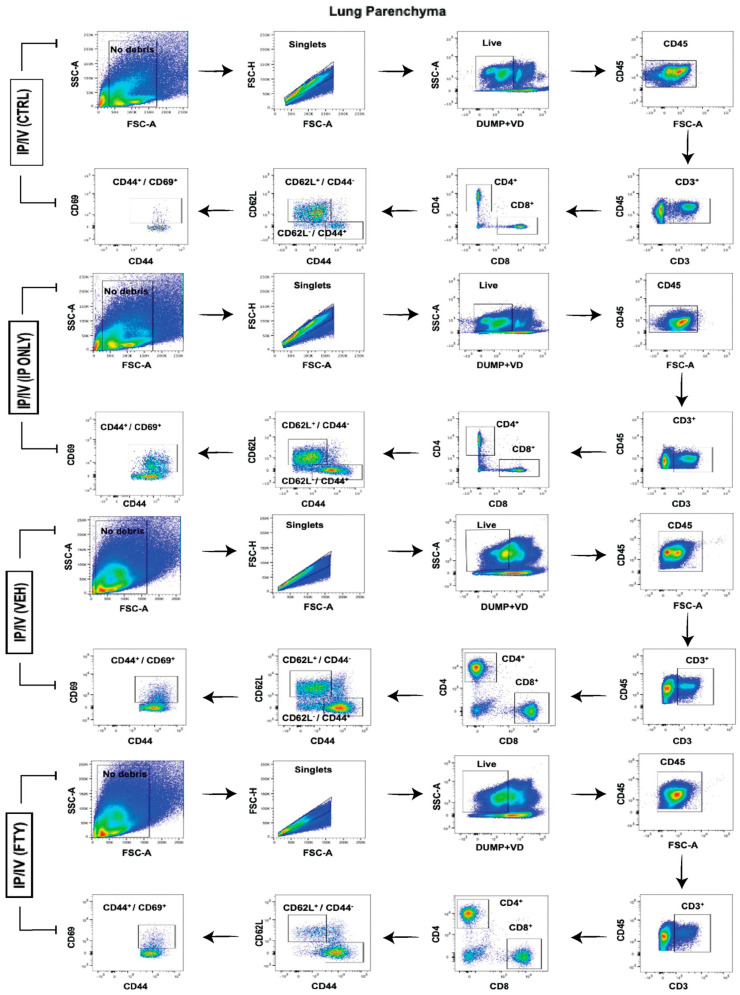
The representative flow cytometry gating strategy of the lung parenchyma for each condition.

**Figure 6 ijms-25-09202-f006:**
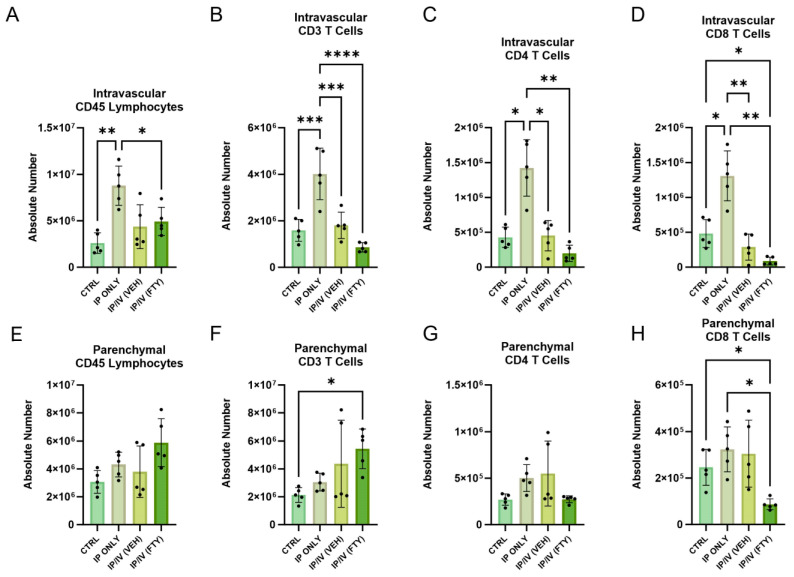
In IP/IV mice, FTY720 treatment decreased CD8 T cell density, but otherwise did not significantly alter lymphocytes and CD4 T cells in the lung parenchyma. The absolute number of intravascular/extraparenchymal (**A**) CD45+ lymphocytes, (**B**) CD3+ T cells, (**C**) CD4+ T cells, and (**D**) CD8+ T cells per lung. The absolute number of extravascular/intraparenchymal (**E**) CD45+ lymphocytes, (**F**) CD3+ T cells, (**G**) CD4+ T cells, and (**H**) CD8+ T cells per lung. Mean +/− SD plotted; N = 6–8/group; ANOVA with post hoc Tukey testing *p* values shown; * *p* < 0.05, ** *p* < 0.01, *** *p* < 0.001, **** *p* < 0.0001.

**Figure 7 ijms-25-09202-f007:**
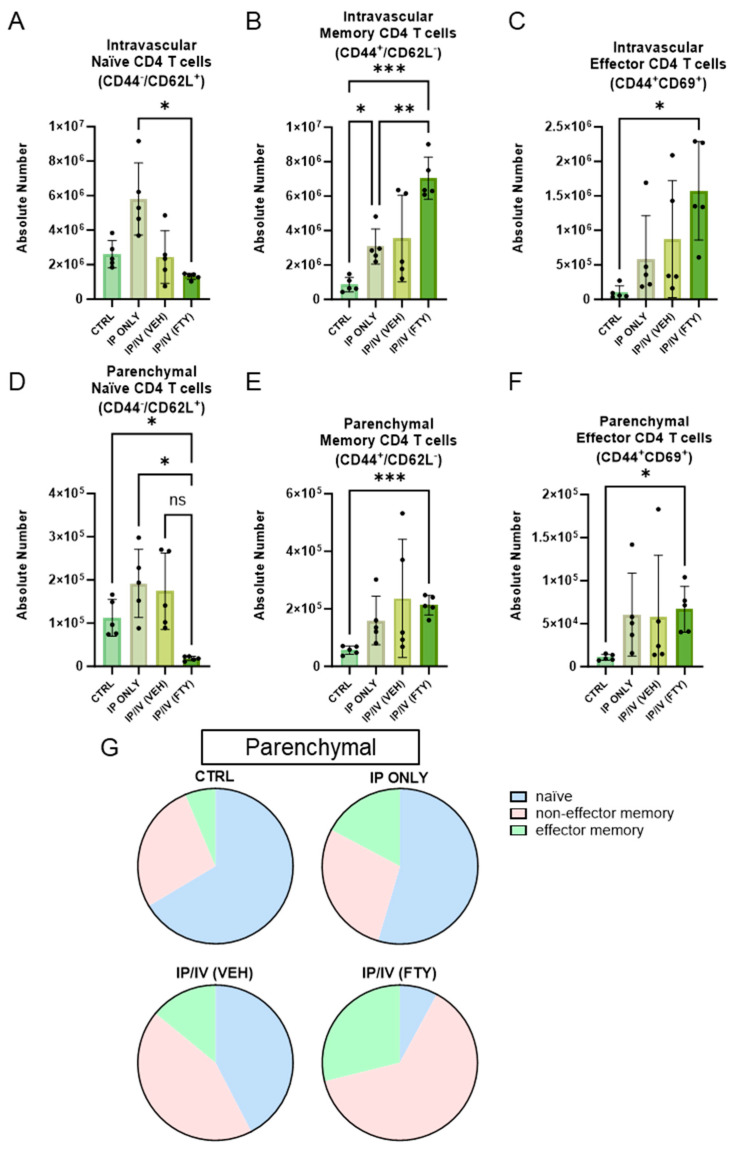
FTY720 treatment significantly decreased the number of naïve CD4 T cells, but did not significantly impact the number of memory or effector lymphocytes in the lung parenchyma. The absolute number of intravascular (**A**) naïve (CD62L^+^CD44^−^), (**B**) memory (CD62L^−^CD44^+^), and (**C**) effector (CD62L^−^CD44^+^CD69^+^) cells in the lung parenchyma, per lung. The absolute number of parenchymal (**D**) naïve, (**E**) memory, and (**F**) effector cells in the lung parenchyma, per lung. Mean +/− SD plotted; N = 5–8/group; ANOVA with post hoc Tukey testing *p* values shown; * *p* < 0.05, ** *p* < 0.01, *** *p* < 0.001. (**G**) The average percentages of parenchymal T cell subsets.

**Table 1 ijms-25-09202-t001:** Antibodies used in flow cytometry.

Antibody Specificity	Fluorochrome	Clone	Final Concentration (μg)	Manufacturer
anti-mouse CD16/32 (blocking Fc domain)	-	93	1	BioLegend
anti-mouse CD19	eFluo450	1D3	0.5	eBioscience
anti-mouse NK1.1	eFluo450	PK136	0.5	eBioscience
anti-mouse Ly6G	eFluo450	1A8	0.5	eBioscience
Fixable Viability dye	eFluo450		0.0625	eBioscience
anti-mouse CD45	AF700	30 F11	1	eBioscience
	BV570	30 F11	2	BioLegend
anti-mouse CD3	FITC	17A2	0.5	BioLegend
anti-mouse CD4	APC	RM4-5	0.5	BioLegend
anti-mouse CD8	PECy7	53-5.8	0.5	BioLegend
anti-mouse CD62L	BV650	MEL-14	0.8	BioLegend
anti-mouse CD45R/B220	BV605	RA3-6B2	1	BioLegend
anti-mouse CD44	PE	IM7	0.5	BioLegend
anti-mouse CD69	BV785	H1.2F3	0.5	BioLegend

## Data Availability

The original contributions presented in the study are included in the article, further inquiries can be directed to the corresponding author/s. The data are available upon reasonable request.

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
