# Peer review of "Intrapulmonary T Cells Are Sufficient for Schistosoma-Induced Pulmonary Hypertension"

_ijms, 2024, doi:10.3390/ijms25179202_

Round 1

Reviewer 1 Report

Comments and Suggestions for Authors

The manuscript by Balladares et al. studies the role of T cells in a model of Schistosoma induced pulmonary hypertension (PH). Understanding the inflammatory processes involved in PH is of much interest and given the prevalence of Schistosoma infection in the developing world is of public health interest. The main question addressed by the current study is if T cell migration from the lymph nodes to the lung parenchyma is required for the development of PH, using a pharmacological blocker of T cell migration, FTY720. Overall, the study was well performed and the manuscript is easy to follow, but I have some questions.

Major questions:

Methods: Details about calculating the Fulton index are given but no data is shown. This data would fit nicely in Figure 3.

Results: Figure 1,2 and 5: Why are the data in Figures 1 and 2 given as percentages when  the main claim is there are more or less cells? Giving absolute numbers would be more infiormative. On the flip side, in Figure 5 the Authors claim that there are changes in cell densities and give absolute cell counts instead of percent values.

The data shown focus on cytokine production and T cell changes. Does FTY treatment have an effect on innate immune cell populations in this model?

Figure 6: In the IP/IV group the data seems to separate into a responding (high effector T cell count) and non-responding group. Did you see similar trends on other inflammatory parameters in these animals (or other experiments using this model?)?

Discussion: A significant portion of the discussion reads (eg. lines 247-278) read more like a literature review than a discussion of the current results.

Minor comments:

Line 28: Please recheck this sentence, there seems to be a word missing

Methods: Am I correct in understanding that the experiments performed on 6 week old mice? If yes, is there a reason for using such young animals?

Anti-SMA staining: Please add dilution information for this antibody, just like for the flow antibodies.

Line 104: Please add details about the protease inhibitors used.

Table 1: For some antibodies, there are no units given.

Line 205: The changes in FTY treated animals are quite large for all T cell populations shown, not modest in my opinion. Please rephrase.

Regarding the limitations of the pharmacological approach: Was the role of T cell migration in PH tested using KO animals, eg. A S1PR KO line?

Author Response

Comments 1: The manuscript by Balladares et al. studies the role of T cells in a model of Schistosoma induced pulmonary hypertension (PH). Understanding the inflammatory processes involved in PH is of much interest and given the prevalence of Schistosoma infection in the developing world is of public health interest. The main question addressed by the current study is if T cell migration from the lymph nodes to the lung parenchyma is required for the development of PH, using a pharmacological blocker of T cell migration, FTY720. Overall, the study was well performed and the manuscript is easy to follow, but I have some questions.

Response 1: Thank you for the positive comments.

Comments 2: Major questions:

Methods: Details about calculating the Fulton index are given but no data is shown. This data would fit nicely in Figure 3.

Response 2: Thank you for this suggestion. The Fulton index data have now been added in Figure 3 as panel C, as requested by the reviewer, and commented upon in the Results section. Note that there was no significant difference across groups: we attribute this to the relatively short duration of pulmonary hypertension (7 days after IV egg administration).

Comments 3: Results: Figure 1, 2 and 5: Why are the data in Figures 1 and 2 given as percentages when the main claim is there are more or less cells? Giving absolute numbers would be more informative. On the flip side, in Figure 5 the Authors claim that there are changes in cell densities and give absolute cell counts instead of percent values.

Response 3: Thank you for this comment. We are unfortunately unable to provide absolute cell counts for the peripheral blood, as the volume of blood drawn was not recorded at the time of the experiment; we only have percentages as was previously shown in Figure 1. Please note that FTY-720 is not expected to significantly affect other circulating cell types, so the use of total lymphocytes as a denominator should reflect changes in T cell density.

Figure 2 displaying mediastinal lymph node data has been revised to now been revised to include both absolute values (total cells per lymph node) as well as percentages (fraction of CD45+ lymphocytes). A limitation of this approach is that the lymph node volume or mass could be affected by FTY-720 treatment; this is mentioned as a limitation in the results section.

Figure 5 has been significantly revised. The new Figure 5 shows the flow cytometry gating approach. A new Figure 6 has been added which shows absolute numbers of cell densities in the lung parenchyma, as well as in the intravascular space to address a comment by Reviewer 2. The absolute numbers are cells per lung.

Comments 4: The data shown focus on cytokine production and T cell changes. Does FTY treatment have an effect on innate immune cell populations in this model?

Response 4: Thank you for this comment. We did not analyze impacts on innate immune cells in this work, as trafficking of these cells is not regulated by S1PR, the protein targeted by FTY-720 treatment. Our current working model for schistosomiasis pulmonary hypertension pathogenesis is that the proximate Th2 adaptive immunity stimulates the recruitment of monocytes to become interstitial macrophages. As we observed no change in the degree of Th2 immunity (IL-4/IL-13 concentrations were unchanged), and no change in the degree of pulmonary hypertension severity, we would reasonably anticipate that monocyte recruitment as the critical intermediary would also be unchanged. This comment has been added to the discussion section in the manuscript.

Comments 5: Figure 6: In the IP/IV group the data seems to separate into a responding (high effector T cell count) and non-responding group. Did you see similar trends on other inflammatory parameters in these animals (or other experiments using this model?)?

Response 5: Thank you for this comment. No, we generally don’t see grouping of these inbred mice into different phenotypes. There is likely some mouse-to-mouse variability in the number of eggs that were administered, clumping of the eggs within the vessels or more uniform distribution, and sampling of the lung tissue that results in more or less inflamed areas of lung being selected for analysis. A comment discussing this point has been added to the results section.

Comments 6: Discussion: A significant portion of the discussion reads (eg. lines 247-278) read more like a literature review than a discussion of the current results.

Response 6: Thank you for this comment; this section has been revised to more clearly reflect the interpretation of our data in the context of the existing literature. In particular, we contract our results to hypoxia-induced PH, which has a similar requirement for CD4 T cells, but here of a Th17 phenotype rather than a Th2 phenotype as we observe in Schistosoma-induced PH. We also summarize our understanding of the current working model for how Schistosoma-PH is induced, particularly in light of our findings in the present work; to our knowledge this summation has not been previously presented elsewhere.

Comments 7: Minor comments:

Line 28: Please recheck this sentence, there seems to be a word missing

Response 7: Thank you for this comment, the word “in” has been added.

Comments 8: Methods: Am I correct in understanding that the experiments performed on 6 week old mice? If yes, is there a reason for using such young animals?

Response 8: Thank you for this comment. We have typically used 6 to 8 week old mice at the start of the experiment; after the 3 week IP sensitization and IV challenge protocol the resulting mice are 9 to 11 weeks at the conclusion of the experiment. We acknowledge this is an age that is equivalent to late adolesence/early adulthood. This age has historically been used to induce a robust immune phenotype. Future experiments are planned to compare the phenotype in younger versus aged mice. A comment on this limitation has been added to the discussion section.

Comments 9: Anti-SMA staining: Please add dilution information for this antibody, just like for the flow antibodies.

Response 9: Thank you for this comment; the immunostaining dilution factor (1:200) has now been added.

Comments 10: Line 104: Please add details about the protease inhibitors used.

Response 10: Thank you for this comment; details about the protease inhibitors used have now been added.

Comments 11: Table 1: For some antibodies, there are no units given.

Response 11: Thank you for this comment; all units have now been added.

Comments 12: Line 205: The changes in FTY treated animals are quite large for all T cell populations shown, not modest in my opinion. Please rephrase.

Response 12: Thank you for this comment; the phrase has been revised to “significant” to reflect the statistical significance of the difference.

Comments 13: Regarding the limitations of the pharmacological approach: Was the role of T cell migration in PH tested using KO animals, eg. A S1PR KO line?

Response 13: Thank you for this comment; a S1PRKO line was not tested, but would be an interesting comparator; this limitation has been noted in the discussion section.

Reviewer 2 Report

Comments and Suggestions for Authors

Comments:

In the paper “Intrapulmonary T Cells are Sufficient for Schistosoma-induced Pulmonary Hypertension” by Balladares et al. the authors block the lymphocyte egress from lymph nodes   and measures the total T cell number by flow cytometry, PH severity by heart catheterization, and cytokine 22 concentration by ELISA. However, FTY720 treatment caused no change in PH or 24 type 2 inflammation severity in mice sensitized and challenged with Schistosoma eggs, and the 25 number of memory and effector CD4 T cells in the lung parenchyma was also unchanged. The experiments performed by the authors look straight forward with proper control and the observation matches the prior data too. But there is some concern that need to be addressed before the paper get accepted.

Major Revision:

·         The work lacks novelty and most of the observation made by the authors are already reported earlier. It will be interesting if the author can explain how their work is different from others and what new observation we can derive from this work will be exciting.

·         In Fig 5 the author quantifies the resident T cell, but the gating strategy has some issues. Resident T-cells are IV negative (CD45) the author should inject the animals with CD45 (IV) and gate the T cells on the IV negative cells. The authors can refer some published paper to correct the getting strategy.

Author Response

Comments 1: In the paper “Intrapulmonary T Cells are Sufficient for Schistosoma-induced Pulmonary Hypertension” by Balladares et al. the authors block the lymphocyte egress from lymph nodes   and measures the total T cell number by flow cytometry, PH severity by heart catheterization, and cytokine 22 concentration by ELISA. However, FTY720 treatment caused no change in PH or 24 type 2 inflammation severity in mice sensitized and challenged with Schistosoma eggs, and the 25 number of memory and effector CD4 T cells in the lung parenchyma was also unchanged. The experiments performed by the authors look straight forward with proper control and the observation matches the prior data too. But there is some concern that need to be addressed before the paper get accepted.

Response 1: Thank you for the comments: we appreciate your review. We have attempted to address the concerns you raised below.

Comments 2: Major Revision:

The work lacks novelty and most of the observation made by the authors are already reported earlier. It will be interesting if the author can explain how their work is different from others and what new observation we can derive from this work will be exciting.

Response 2: Thank you for the comment. Respectfully, we submit that this work studies an important step in the pathogenesis of schistosomiasis pulmonary hypertension, namely the activation of CD4 T cells and where they are located. Prior to this work, we had suspected that primed CD4 T cells would be recruited to the lung tissue upon administration of intravenous eggs; we were thus surprised that CD4 T cells already positioned in the lung tissue were sufficient to induce the Type 2 immunity necessary for Schistosoma-induced PH to occur. This data is concordant with reports from other infectious diseases, as reviewed in our discussion section, but has not been investigated in the context of pulmonary hypertension before to our knowledge. We also draw important comparisons to the widely used hypoxia model of experimental PH. These points has been clarified in the revised discussion section.

Comments 3: In Fig 5 the author quantifies the resident T cell, but the gating strategy has some issues. Resident T-cells are IV negative (CD45) the author should inject the animals with CD45 (IV) and gate the T cells on the IV negative cells. The authors can refer some published paper to correct the getting strategy.

Response 3: Thank you for the comment. Intravascular anti-CD45 was administered prior to sacrifice, enabling distinction between the intravascular/extraparenchymal and extravascular/intraparenchymal cells. We have now revised our results in Figures 5, 6 and 7 to reflect the distinction between these two populations, and revised our interpretation of the results appropriately. In particular, IP sensitization alone increased the density of intravascular CD45 lymphocytes and CD4 T cells, and specifically naïve and memory CD4 T cells. In contrast, FTY720 treatment significantly decreased the number of intraparenchymal naïve CD4 T cells. These important results allow us to further refine our interpretation of our data, but importantly do not contradict any of our prior findings.

Reviewer 3 Report

Comments and Suggestions for Authors

In the prsent study by Dara C. Fonseca Balladares et al, authors have investigated the role of resident T cells in pulmonary hypertension following Schistosoma egg challenge.

Authors have wittern the manuscript very well. Methodology, results are properly written and concluded appropriately.

The paper can be accepted in the current form. 

Author Response

Comments 1: In the present study by Dara C. Fonseca Balladares et al, authors have investigated the role of resident T cells in pulmonary hypertension following Schistosoma egg challenge.

Authors have written the manuscript very well. Methodology, results are properly written and concluded appropriately.

The paper can be accepted in the current form. 

Response 1: Thank you for the positive comments.

Round 2

Reviewer 1 Report

Comments and Suggestions for Authors

Thank you for addressing my comments. I recommend acceptance.

Reviewer 2 Report

Comments and Suggestions for Authors

The authors clearly addressed all the comments in the modified manuscript.